# Chaos and Thermalization in the Spin-Boson Dicke Model

**DOI:** 10.3390/e25010008

**Published:** 2022-12-21

**Authors:** David Villaseñor, Saúl Pilatowsky-Cameo, Miguel A. Bastarrachea-Magnani, Sergio Lerma-Hernández, Lea F. Santos, Jorge G. Hirsch

**Affiliations:** 1Instituto de Ciencias Nucleares, Universidad Nacional Autónoma de México, Apdo. Postal 70-543, Mexico City 04510, Mexico; 2Instituto de Investigaciones en Matemáticas Aplicadas y en Sistemas, Universidad Nacional Autónoma de México, Mexico City 04510, Mexico; 3Center for Theoretical Physics, Massachusetts Institute of Technology, Cambridge, MA 02139, USA; 4Departamento de Física, Universidad Autónoma Metropolitana-Iztapalapa, Av. Ferrocarril San Rafael Atlixco 186, Mexico City 09340, Mexico; 5Facultad de Física, Universidad Veracruzana, Circuito Aguirre Beltrán s/n, Xalapa 91000, Mexico; 6Department of Physics, University of Connecticut, Storrs, CT 06269, USA

**Keywords:** quantum chaos, eigenstate thermalization hypothesis, quantum entanglement

## Abstract

We present a detailed analysis of the connection between chaos and the onset of thermalization in the spin-boson Dicke model. This system has a well-defined classical limit with two degrees of freedom, and it presents both regular and chaotic regions. Our studies of the eigenstate expectation values and the distributions of the off-diagonal elements of the number of photons and the number of excited atoms validate the diagonal and off-diagonal eigenstate thermalization hypothesis (ETH) in the chaotic region, thus ensuring thermalization. The validity of the ETH reflects the chaotic structure of the eigenstates, which we corroborate using the von Neumann entanglement entropy and the Shannon entropy. Our results for the Shannon entropy also make evident the advantages of the so-called “efficient basis” over the widespread employed Fock basis when investigating the unbounded spectrum of the Dicke model. The efficient basis gives us access to a larger number of converged states than what can be reached with the Fock basis.

## 1. Introduction

The onset of thermalization in isolated quantum systems, which evolve unitarily and are described by pure states, was discussed in von Neumann’s 1929 paper on the quantum ergodic theorem [1,2,3,4]. In Pechukas’s 1984 paper “Remarks on Quantum Chaos” [5], one finds a brief summary of von Neumann’s work and the subsequent developments against it, particularly by Bocchieri and Loinger [6], which, Pechukas claims, actually make von Neumann’s results sharper [7]. In Pechukas’s words,

if one selects a state “at random” from an energy shell and determines, as a function of time, the probability that it lies in a “typical” subspace of the shell, the time average of this probability is liable to be much closer to the statistical expectation than is its instantaneous value, the more so the less degenerate the spectrum of the Hamiltonian.

The ideas in this quotation are connected with the notion of “typicality” [3,4], which has been one of the directions in studies of thermalization.

In retrospect, one can identify in the 1985 work by Jensen and Shakar [8] the seeds for what later became known as the eigenstate thermalization hypothesis (ETH) [9,10]. There, the authors study the infinite-time average and the eigenstate expectation value of the magnetization of a finite spin-1/2 chain for different initial states and different energy eigenstates, and show that both exhibit very good agreement with the microcanonical average when the system is chaotic. They state that in the chaotic system, “the magnetization is a fairly smooth and monotonic function of energy”, and in this case,

even an initial energy eigenstate will exhibit a constant value for the observable which is very close to that predicted by the statistical theory,

while for the integrable Hamiltonian, the “deviations from equilibrium tend to be large”. This means that an observable *O* should thermalize when its expectation value $\langle E_{k}|\hat{O}|E_{k}\rangle$ is a smooth function of energy, since in this situation, its infinite-time average, $\bar{O}$, will be very close to the microcanonical average, $O_{\mathrm{mic}}$; that is [11]
(1)$$\bar{O}=\sum_{k}{\left|c_{k}\right|}^{2}O_{k,k}≃O_{\mathrm{mic}}=\frac{1}{W_{E,\Delta E}}\sum_{k}O_{k,k}\;\;,$$
where $|E_{k}\rangle$ are the eigenstates of the system’s Hamiltonian, $O_{k,k}=\langle E_{k}|\hat{O}|E_{k}\rangle$ are the diagonal elements of the operator $\hat{O}$, $c_{k}=\langle E_{k}|\Psi \left(0\right)\rangle$ are the coefficients of the initial state $|\Psi \left(0\right)\rangle =\sum_{k}c_{k}|E_{k}\rangle$, and $W_{E,\Delta E}$ is the number of eigenstates $|E_{k}\rangle$ contained in the energy window $E_{k}\in \left[E-\Delta E,E+\Delta E\right]$ with $|E-E_{k}|<\Delta E$. Under the assumption of the smooth behavior in energy, even a single eigenstate inside the microcanonical window should give $O_{k,k}$ very close to the microcanonical average; therefore, the term “eigenstate thermalization hypothesis” (ETH) was coined by Srednicki in his 1994 paper [12]. Notions of quantum chaos [8], random matrices [13], and Berry’s conjecture [12] have been invoked to justify the validity of Equation (Equation 1). Starting with Rigol et al’s 2008 paper [11], several numerical studies have confirmed Equation (Equation 1) for chaotic systems. Since then, various studies further elaborated the framework of the ETH to take into account the convergence of $\bar{O}$ and $O_{\mathrm{mic}}$ as the system size increases [9,14], the analysis of the off-diagonal elements $O_{k,k^{\prime }}=\langle E_{k}|\hat{O}|E_{k^{\prime }}\rangle$, known as off-diagonal ETH [9,15], the dependence on the energy of the initial state [16,17], and the structure of the eigenstates in realistic many-body quantum systems [18,19,20].

The analysis in Ref. [18] was inspired by the 1990s papers from previous members of the Novosibirsk school, including Casati [21], Flambaum [22,23], Izrailev [24,25], Shepelyansky [26], and Zelevinsky [27,28], who focused on the structure of the eigenstates to define quantum chaos and explain the onset of thermalization [29]. In realistic interacting many-body quantum systems, the energy eigenstates can be rather complicated, but they are not random vectors, as the eigenstates of Gaussian random matrices or the random superpositions of plane waves with random phases and Gaussian random amplitude stated by the Berry’s conjecture [30]. For such random vectors, Equation (Equation 1) is trivially satisfied, but they are associated with unphysical models. In the above mentioned 1990s papers, chaotic eigenstates are those that, when written in the mean field basis, exhibit coefficients that are random variables following a Gaussian distribution around the envelope defined by the energy shell. They emerge away from the edges of the spectrum of quantum systems with many strongly interacting particles and ensure the validity of the Fermi–Dirac [24] and Bose–Einstein [31] distributions. These ideas were employed in Ref. [18] to corroborate the validity of the ETH for few-body observables when the eigenstates are chaotic.

The present article is dedicated to Professor Giulio Casati on the occasion of his 80th birthday in 2022 and to Professor Felix Izrailev’s 80th birthday in 2021. Not only their past, but also their recent works [31,32,33,34,35,36,37,38] continue to have an enormous impact in the developments of quantum chaos and its connections with thermalization, quantum statistical mechanics, and the quantum-classical correspondence.

Here, we investigate the validity of the diagonal and off-diagonal ETH, in connection with the structure of the eigenstates, for the spin-boson Dicke model [39,40,41]. This system has a well-defined classical limit and two-degrees of freedom, so it does not quite fall within the requirement for thermalization of a large number of coupled degrees of freedom [13].

Depending on the parameters and energy region, the Dicke model may be regular or chaotic. Since its introduction in the 1950s to explain superradiance in spontaneous radiation processes [39], it has been employed in a variety of theoretical studies that include quantum phase transitions [42,43,44], classical and quantum chaos [45,46,47,48,49], non-equilibrium quantum dynamics [41,50,51,52,53,54], the evolution of out-of-time-ordered correlators (OTOCs) [55,56,57], quantum scarring [58,59,60,61,62], and quantum localization measures in phase space [61,63]. Experimentally, the model can be realized with superconducting circuits [64], cavity assisted Raman transitions [65,66], trapped ions [67,68], and other systems [69]. In the particular context of the relationship between chaos and thermalization, the Dicke model was studied in Ref. [70] with emphasis on the behavior of the fidelity OTOC and the agreement between the long-time average of the collective spin observable and the microcanonical ensemble. The chaos–thermalization connection was also explored for the kicked Dicke model in Ref. [71].

We analyze the diagonal and off-diagonal ETH for the number of photons and the number of excited atoms in the regular and chaotic regions of the Dicke model, and compare the results with the structure of the eigenstates analyzed with the entanglement entropy and the Shannon entropy. The latter is a delocalization measure that depends on the basis in which the eigenstates are written. The Hilbert space of the Dicke model is infinite, because its number of bosons is unbounded. Our results show that the values of the Shannon entropy grows rapidly with the eigenvalues when the eigenstates are written in the Fock basis, but are more restricted when the “efficient basis” is used, which makes evident the advantages of the latter when one wants to study large systems and high energies.

The paper is organized as follows. We introduce the Dicke model in Section 2 and briefly review the onset of classical and quantum chaos in Section 3. In Section 4, we study the diagonal and off-diagonal ETH and analyze the eigenstates in Section 5. Our conclusions are summarized in Section 6.

## 2. Dicke Model

The Dicke model [39] describes a system of N two-level atoms coupled with a single mode of a quantized radiation field. Setting ℏ=1, the Hamiltonian is given by
(2)$$\begin{array}{cc} {\hat{H}}_{D} & ={\hat{H}}_{F}+{\hat{H}}_{A}+{\hat{H}}_{I}, \end{array}$$
(3)$$\begin{array}{cc} {\hat{H}}_{F} & =\omega {\hat{a}}^{†}\hat{a}, \end{array}$$
(4)$$\begin{array}{cc} {\hat{H}}_{A} & =\omega_{0}{\hat{J}}_{z}, \end{array}$$
(5)$$\begin{array}{cc} {\hat{H}}_{I} & =\frac{\gamma }{\sqrt{N}}\left({\hat{a}}^{†}+\hat{a}\right)\left({\hat{J}}_{+}+{\hat{J}}_{-}\right), \end{array}$$
where ${\hat{H}}_{F}$ defines the field’s energy, ${\hat{H}}_{A}$ the energy of the two-level atoms, and ${\hat{H}}_{I}$ the atom–field interaction energy. In the equations above, ${\hat{a}}^{†}$ ($\hat{a}$) is the bosonic creation (annihilation) operator of the single field mode and ${\hat{J}}_{+}$ (${\hat{J}}_{-}$) is the raising (lowering) collective pseudo-spin operator, where ${\hat{J}}_{\pm }={\hat{J}}_{x}\pm i{\hat{J}}_{y}$ and ${\hat{J}}_{x,y,z}=\left(1/2\right)\sum_{k=1}^{N}{\hat{\sigma }}_{x,y,z}^{k}$ for the Pauli matrices ${\hat{\sigma }}_{x,y,z}^{k}$ acting on the *k*th two-level atom. Since the squared total pseudo-spin operator ${\hat{J}}^{2}={\hat{J}}_{x}^{2}+{\hat{J}}_{y}^{2}+{\hat{J}}_{z}^{2}$ commutes with the Hamiltonian, $[{\hat{H}}_{D},{\hat{J}}^{2}]=0$, the eigenvalues of ${\hat{J}}^{2}$, given by j(j+1), label the invariant subspaces in the Hilbert space. We work within the totally symmetric subspace, which includes the collective ground state and is defined by the maximum pseudo-spin value j=N/2.

The Dicke Hamiltonian ${\hat{H}}_{D}$ commutes with the parity operator
(6)$$\hat{\Pi }=\exp\left(i\pi \hat{\Lambda }\right),$$
where
(7)$$\hat{\Lambda }={\hat{a}}^{†}\hat{a}+{\hat{J}}_{z}+j\hat{1}=\hat{n}+{\hat{n}}_{\mathrm{ex}},$$
the number of photons is $\hat{n}={\hat{a}}^{†}\hat{a}$, and the number of excited atoms is ${\hat{n}}_{\mathrm{ex}}={\hat{J}}_{z}+j\hat{1}$. Because of this symmetry, the eigenstates have one of two different parities, $\hat{\Pi }|E_{k}\rangle =\pm |E_{k}\rangle$.

The three parameters of the Dicke Hamiltonian, ω, $\omega_{0}$, and γ, determine the energy scales of the system and the onset of chaos. The radiation frequency of the single-mode electromagnetic field (the boson) is given by ω, the energy splitting of each two-level atom is $\omega_{0}$, and γ is the coupling strength modulating the atom–field interaction. In the thermodynamic limit, N→∞, the light-matter coupling divides the parameter space in the normal phase, when the strength γ in smaller than the critical value $\gamma_{c}=\sqrt{\omega \omega_{0}}/2$, and the superradiant phase, when $\gamma >\gamma_{c}$ [42,72,73,74]. The Dicke model also presents regular and chaotic regions depending on the parameters and excitation energies [47]. We fix the coupling strength in the superradiant phase, $\gamma =2\gamma_{c}=1$, and the resonant frequencies at $\omega =\omega_{0}=1$, so that we ensure that both the classical dynamics and the quantum eigenspectrum are chaotic at excitation energies ϵ≥−0.8 [47], where ϵ=E/j denotes the energy scaled to the system size *j*.

The Hilbert space of the Dicke model is infinite. In Appendix A, we provide details on how to diagonalize the system’s Hamiltonian on an efficient basis. We consider three system sizes, j=30,60,100, whose associated Hilbert space dimensions in the efficient basis are given by $d_{D}^{\mathrm{EB}}=8\;601,30\;371,80\;601$, ensuring the convergence of the set of eigenstates from the ground-state energy $\epsilon_{\mathrm{GS}}=-2.125$ until a truncated value $\epsilon_{T}$.

## 3. Chaos in the Dicke Model

The classical Hamiltonian of the Dicke model is presented in Appendix B. The generation of Poincaré sections and the computation of Lyapunov exponents for trajectories in phase space [47,75] reveal the onset of chaos for high energies and strong couplings. In Figure 1a, we show a classical map of the percentage of chaos as a function of excitation energies and coupling strengths. The percentage of chaos is defined as the ratio of the number of chaotic initial conditions over the total number of initial conditions used in the sample and is illustrated with a color gradient, where dark indicates that the majority of the initial conditions are regular and light indicates that most initial conditions are chaotic.

Signatures of the onset of chaos in classical systems are found in the quantum domain, as suggested by Casati et al’s 1980 paper [76] and Bohigas et al’s 1984 work [77], and became known as “quantum chaos”. The eigenvalues of a quantum system become correlated when its classical counterpart is chaotic. The degree of correlations between neighboring levels can be detected with the level spacing distribution [78] or the ratio of consecutive energy levels [79,80],
(8)$$\begin{array}{cc} {\tilde{r}}_{k} & =\min\left(r_{k},r_{k}^{-1}\right)=\frac{\min(s_{k},s_{k-1})}{\max(s_{k},s_{k-1})}, \end{array}$$
where $s_{k}=E_{k+1}-E_{k}$ is the nearest-neighbor spacing between energy levels and $r_{k}=s_{k}/s_{k-1}$; while both short- and long-range correlations are measured with quantities such as the level number variance [81] or the spectral form factor [78]. In the case of the Dicke model, level spacing distributions [75,82,83] and spectral form factors [53,54] in agreement with random matrix theory (RMT) were verified for the energies and parameters associated with the onset of chaos in the classical limit.

Other tests of quantum chaos include Peres lattices [84] and the evolution of OTOCs [85,86,87]. The Peres lattice is a plot of the eigenstate expectation values of an observable as a function of the eigenvalues. It has been widely employed (not always under this name) in studies of ETH and provides visual evidence of the loss of integrability [75,88]. As for the OTOCs, their exponential growth rates are seen as quantum analogs of the classical Lyapunov exponents. This was indeed confirmed for the Dicke model in the chaotic region [55], although its exponential growth happens also in regular regions due to instability [57].

In Figure 1, we compare the onset of classical chaos, investigated in Figure 1a, with the degree of correlations between neighboring energy levels, as captured by the ratio of consecutive energies in Figure 1b. To smooth spectral fluctuations, an average is performed and denoted by $\langle \tilde{r}\rangle$. In the regular regime, where the eigenvalues are uncorrelated and the level spacing distribution is Poissonian, ${\langle \tilde{r}\rangle }_{P}\approx 0.39$, while in the chaotic region, where the eigenvalues are correlated as in RMT and the level spacing distribution follows the Wigner–Dyson distribution, ${\langle \tilde{r}\rangle }_{\mathrm{WD}}\approx 0.53$. Our results in Figure 1 exhibit an evident classical-quantum correspondence. There is a visible relationship between the classical map of Figure 1a and the quantum map of Figure 1b, when the percentage of chaos is large (small) in Figure 1a; $\langle \tilde{r}\rangle$ approaches 0.53 (0.39) in Figure 1b.

## 4. Thermalization in the Dicke Model

For an isolated quantum system initially in a pure state, the evolution is governed by the unitary time operator as
(9)$$|\Psi \left(t\right)\rangle =e^{-i\hat{H}t}|\Psi \left(0\right)\rangle =\sum_{k}c_{k}e^{-iE_{k}t}|E_{k}\rangle ,$$
where $\hat{H}|E_{k}\rangle =E_{k}|E_{k}\rangle$ and $c_{k}=\langle E_{k}|\Psi \left(0\right)\rangle$. Thus, the expectation value of a given operator $\hat{O}$ under states evolved in time can be calculated as follows
(10)$$O\left(t\right)=\langle \Psi \left(t\right)|\hat{O}|\Psi \left(t\right)\rangle =\sum_{k\neq k^{\prime }}c_{k}^{*}c_{k^{\prime }}e^{i(E_{k}-E_{k^{\prime }})t}O_{k,k^{\prime }}+\sum_{k}{\left|c_{k}\right|}^{2}O_{k,k}\;\;\;,$$
where $O_{k,k^{\prime }}=\langle E_{k}|\hat{O}|E_{k^{\prime }}\rangle$ are the matrix elements of the operator expressed in the energy eigenbasis.

The onset of thermalization according to the ETH happens for local observables if two assumptions are satisfied:

(i) The infinite-time average,
(11)$$\bar{O}=\lim_{t\to +\infty }\frac{1}{t}\int_{0}^{t}dt^{\prime }O\left(t^{\prime }\right)=\sum_{k}{\left|c_{k}\right|}^{2}O_{k,k},$$
coincides with the microcanonical average,
(12)$$O_{\mathrm{mic}}=\frac{1}{W_{E,\Delta E}}\sum_{k}O_{k,k}\approx \bar{O},$$
or more precisely, $\bar{O}$ approaches the thermodynamic average as the system size grows. This is sometimes referred to as “diagonal ETH”. In the equation above, $W_{E,\Delta E}$ is the number of eigenstates $|E_{k}\rangle$ contained in the energy window $E_{k}\in \left[E-\Delta E,E+\Delta E\right]$ with $|E-E_{k}|<\Delta E$.

As explained above, the ETH should be valid when the eigenstates are chaotic (ergodic), filling the energy shell. In this case, one obtains Gaussian distributions for few-body observables in many-body quantum systems, which can be understood as follows. Assume that the few-body observable $\hat{O}$ has *N* nonzero eigenvalues $O_{n}$ in $\hat{O}|O_{n}\rangle =O_{n}|O_{n}\rangle$, where *N* is large. We can project the energy eigenstates in the basis $|O_{n}\rangle$ as $|E_{k}\rangle =\sum_{n}C_{O_{n}}^{k}|O_{n}\rangle$, and write
(13)$$O_{kk}=\langle E_{k}|\hat{O}|E_{k}\rangle =\sum_{n=1}^{N}|C_{O_{n}}^{k}{|}^{2}\langle O_{n}|\hat{O}|O_{n}\rangle =O_{1,1}|C_{O_{1}}^{k}{|}^{2}+\ldots +O_{N,N}{\left|C_{O_{N}}^{k}\right|}^{2}.$$

If the eigenstates are fully chaotic (ergodic), then $C_{O_{n}}^{k}$’s are independent Gaussian random numbers. According to the central limit theorem, a large sum of independent random numbers $O_{n,n}{\left|C_{O_{n}}^{k}\right|}^{2}$ will also follow a Gaussian distribution. Therefore, in the region where the eigenstates are chaotic, the distribution of $O_{k,k}$ should be Gaussian.

(ii) The fluctuations around $\bar{O}$, which are determined by the phases $e^{i(E_{k}-E_{k^{\prime }})t}$, the coefficients $c_{k}$, and the off-diagonal elements $O_{k,k^{\prime }}$ in Equation (Equation 10), decrease with system size and cancel out on average. This is sometimes referred to as “off-diagonal ETH”. A Gaussian distribution of $O_{k,k^{\prime }}$ indicates that the eigenstates are strongly chaotic (ergodic), so condition (ii) should be satisfied if the energy of the initial state (of a system perturbed far from equilibrium) is in the chaotic region.

The relationship between the Gaussian distribution of the off-diagonal elements of a few-body observable and ergodicity can be understood as above, using $\hat{O}|O_{n}\rangle =O_{n}|O_{n}\rangle$ and $|E_{k}\rangle =\sum_{n}C_{O_{n}}^{k}|O_{n}\rangle$ in
(14)$$O_{kk^{\prime }}=\langle E_{k}|\hat{O}|E_{k^{\prime }}\rangle =\sum_{n=1}^{N}{\left(C_{O_{n}}^{k}\right)}^{*}C_{O_{n}}^{k^{\prime }}\langle O_{n}|\hat{O}|O_{n}\rangle =O_{1,1}{\left(C_{O_{1}}^{k}\right)}^{*}C_{O_{1}}^{k^{\prime }}+\ldots +O_{N,N}{\left(C_{O_{N}}^{k}\right)}^{*}C_{O_{N}}^{k^{\prime }}.$$

For chaotic eigenstates, $C_{O_{n}}^{k}$’s are independent Gaussian random numbers, and the product of two independent Gaussian random numbers, ${\left(C_{O_{n}}^{k}\right)}^{*}C_{O_{n}}^{k^{\prime }}$, which appears in each term of Equation (Equation 14), is also an independent random number. According to the central limit theorem, a large sum of independent random numbers follows a Gaussian distribution, so in the region where the eigenstates are chaotic, the distribution of $O_{k,k^{\prime }}$ should be Gaussian. This has been confirmed for different chaotic systems with many degrees of freedom [15,89,90], but not in chaotic systems with one [91] or few particles [92], few degrees of freedom [93] or in many-body systems when $\hat{O}$ is not few-body [94].

### 4.1. Diagonal ETH

To test the validity of the diagonal ETH, we compute the deviation of the eigenstate expectation values with respect to the microcanonical value [19,20]
(15)$$\Delta^{\mathrm{mic}}\left[O\right]=\frac{\sum_{k}\left|O_{k,k}-O_{\mathrm{mic}}\right|}{\sum_{k}O_{k,k}},$$
and also use a stronger test that takes into account the normalized extremal fluctuations and is given by [19]
(16)$$\Delta_{e}^{\mathrm{mic}}\left[O\right]=\left|\frac{\max(O)-\min(O)}{O_{\mathrm{mic}}}\right|,$$
where max(O) and min(O) are taken from the same energy window $E_{k}\in \left[E-\Delta E,E+\Delta E\right]$ used in Equation (Equation 12). We consider as observables the number of excited atoms, ${\hat{n}}_{\mathrm{ex}}={\hat{J}}_{z}+j\hat{1}$, and the number of photons, $\hat{n}={\hat{a}}^{†}\hat{a}$, of the Dicke model.

In Figure 2, we show the Peres lattices for the expectation values of the number of photons, $n_{k,k}=\langle E_{k}|\hat{n}|E_{k}\rangle$ (Figure 2a), and of the number of excited atoms, ${\left(n_{\mathrm{ex}}\right)}_{k,k}=\langle E_{k}|{\hat{n}}_{\mathrm{ex}}|E_{k}\rangle$ (Figure 2b), for all eigenstates of the Dicke model with positive parity that range from the ground state energy, $\epsilon_{\mathrm{GS}}=-2.125$, up to a maximal converged eigenstate with eigenenergy $\epsilon_{T}=1.755$. In the regular region of low energies, the Peres lattices present a clear pattern related with the quasi-conserved quantities. Above ϵ≈−0.8, where the system becomes chaotic, the lattices become smoother in energy. It is visible that the spread of the expectation values in the chaotic region decreases as the system size increases; that is, for high excitation energies, the fluctuations are larger for j=30 than for j=100. It is also noticeable that the number of excited atoms fluctuate less than the number of photons.

The reduction of the fluctuations with the increase of system size, which is a necessary condition for the validity of the ETH, is better quantified in Figure 2c,d, where we show $\Delta^{\mathrm{mic}}\left[n\right]$ and $\Delta^{\mathrm{mic}}\left[n_{\mathrm{ex}}\right]$, respectively. To compute these quantities, we used moving windows of eigenenergies for each system size j=30,60,100, which contain 100,350,900 energy levels. In the chaotic region, both deviations clearly decrease as *j* increases, while at low energies (ϵ<−0.8), where chaos and regularity coexist, the fluctuations decrease very slowly or do not decrease at all.

The insets of Figure 2c,d contain the results for the normalized extremal fluctuation in Equation (Equation 16). Only the chaotic energy interval defined by ϵ∈[−0.5,1.755] is shown. The reduction of the extremal fluctuations as *j* increases is also perceptible in these insets, thus confirming the validity of the diagonal ETH in the chaotic region of the Dicke model.

In Figure 3, we consider the energy interval ϵ∈[0.5,1], where hard chaos manifests itself classically, and use only the positive parity sector of eigenstates. In Figure 3a,b, we show, respectively, the distribution of the diagonal elements of the number of photons and the number of excited atom, which are presented in a semi-logarithmic scale. The figures confirm that the distribution shape is Gaussian, as explained in Equation (Equation 13).

### 4.2. Off-Diagonal ETH

We now analyze the off-diagonal elements of the number of photons in Figure 3c and the number of excited atoms in Figure 3d. As discussed in Equation (Equation 14), a Gaussian distribution validates the off-diagonal ETH. This is indeed the case of Figure 3c,d.

## 5. Entropies of the Eigenstates of the Dicke Model

In the last section, we verified numerically the validity of the ETH in the chaotic region of the Dicke model using the matrix elements of the number of photons and the number of excited atoms. In this section, we use the von Neumann entanglement entropy and the Shannon entropy to analyze the structure of the eigenstates of the model. The von Neumann entanglement entropy can be regarded as the limit of higher-order observables in a replicated Hilbert space, and one may generalize the ETH to include higher-order statistical moments [95] arising from these replicated spaces, which allows one to explain the so-called Page correction [96] to the volume law. The von Neumann entanglement entropy has been linked to the onset of chaos in quantum systems [97,98,99,100,101]. The Shannon entropy is a basis-dependent quantity. We study this entropy with respect to both the Fock and an efficient basis.

The Dicke model is a bipartite system, whose Hilbert space is a tensor product of the atomic $H_{A}$ and bosonic $H_{B}$ sectors, $H_{D}=H_{A}⊗H_{B}$. The bosonic sector is infinite-dimensional, while the atomic one has dimension $d_{A}=2j+1$. For a pure state expanded in the Fock basis $|n;j,m_{z}\rangle$,
(17)$$|\Psi \rangle =\sum_{n=0}^{\infty }\sum_{m_{z}=-j}^{j}c_{n,m_{z}}|n;j,m_{z}\rangle ,$$
the density matrix is the projector operator $\hat{\rho }=\left|\Psi \rangle \langle \Psi \right|$, and the reduced density matrix in the atomic sector is calculated as,
(18)$$\begin{array}{c} {\hat{\rho }}_{A}={\mathrm{Tr}}_{B}\left[\hat{\rho }\right]=\sum_{m_{z}=-j}^{j}\sum_{m_{z}^{\prime }=-j}^{j}\left(\sum_{n=0}^{\infty }c_{n,m_{z}}c_{n,m_{z}^{\prime }}^{*}\right)|j,m_{z}\rangle \langle j,m_{z}^{\prime }|. \end{array}$$

The von Neumann entanglement entropy is given by
(19)$$\begin{array}{cc} S_{\mathrm{En}}= & -\mathrm{Tr}[{\hat{\rho }}_{A}\ln\left({\hat{\rho }}_{A}\right)]. \end{array}$$

For numerical convenience, we trace out the infinite bosonic sector first, but from the Schmidt decomposition [102], the result for $S_{\mathrm{En}}$ is the same if we would instead trace out the atomic sector first. (See Appendix C for a generalization of quantum entanglement to multipartite systems.)

The standard basis to compute the entanglement entropy is the Fock basis, which is defined as a decoupled basis between photons and atoms $|n;j,m_{z}\rangle =|n\rangle ⊗|j,m_{z}\rangle$ (see Section A.1). Nevertheless, alternative bases can be used to calculate this entropy by selecting the correct partition of the system. In this work, in addition to the Fock basis, we use an efficient basis (see Section A.2 for a full description) that allows us to reach larger system sizes (j>30) than we can reach with the Fock basis [103,104]. The efficient basis is defined as the tensor product $|N;j,m_{x}{\rangle =|N\rangle }_{m_{x}}⊗|j,m_{x}\rangle$, where $|j,m_{x}\rangle$ are the atomic pseudo-spin states rotated by an angle −π/2 around the *y* axis and ${|N\rangle }_{m_{x}}$ are displaced Fock states, whose displacing depends explicitly on the atomic pseudo-spin eigenvalue $m_{x}$. In the following general notation for a pure state,
(20)$$|\Psi \rangle =\sum_{x=0}^{\infty }\sum_{y=-j}^{j}c_{x,y}|x;j,y\rangle ,$$
$\left(x,y\right)=\left(n,m_{z}\right)$ stands for the Fock basis and $\left(x,y\right)=\left(N,m_{x}\right)$ for the efficient basis.

To select the correct atomic sector starting with the efficient basis, we need to perform an adequate trace over the modified bosonic sector related with this basis, such that it is equivalent to the trace over the bosonic sector of the Fock basis. This can be accomplished by mapping one basis into the other (see Section A.3). As the Hilbert space associated with the atomic pseudo-spin states $|j,m_{z}\rangle$ is the same as the Hilbert space of the rotated ones $|j,m_{x}\rangle$, the atomic sector for both subspaces is the same and the entanglement entropy can be computed properly. The Shannon entropy of the pure state in Equation (Equation 20) is given by
(21)$$S_{\mathrm{Sh}}=-\sum_{x=0}^{\infty }\sum_{y=-j}^{j}|c_{x,y}{|}^{2}\ln(|c_{x,y}{|}^{2}).$$

### 5.1. Results for the Entanglement Entropy

We start by comparing the von Neumann entanglement entropy for the Dicke model, which is nonintegrable, with the results for one of the integrable limits of the model, namely the Tavis–Cummings model (see Appendix D for the derivation of this model). This comparison was also made in Ref. [105].

In Figure 4, we plot the Peres lattice of the exponential of the von Neumann entanglement entropy in Equation (Equation 19) for the integrable Tavis–Cummings model (Figure 4a) and the nonintegrable Dicke model (Figure 4b). Results for all states from both parity sectors, from the ground-state of the Tavis–Cummings model up to a maximal converged eigenstate with eigenenergies $\epsilon_{k}\in \left[\epsilon_{\mathrm{GS}}=-2.136,\epsilon_{T}=2.5\right]$, are shown. As evident in Figure 4a, the values of the entropy for the integrable case show large fluctuations and patterns associated with regularity are visible. The fluctuations indicate that even states very close in energy may have very different structures, so ETH should not be satisfied. In contrast, $S_{\mathrm{En}}$ becomes a smoother function of energy in the chaotic region of the Dicke model (ϵ>−0.8), as seen in Figure 4b, which indicates the states close in energy are very similar. In the case of the Dicke model, regular patterns are restricted to the low energies, where the model is not chaotic.

We stress, however, that some isolated eigenstates located in the chaotic regime of the Dicke model present low entanglement. They should be related with strongly scarred states, which are known to exist in this model [61,62,106].

In Figure 5a, we analyze the von Neumann entanglement entropy for the eigenstates of the Dicke model, ranging from the ground state until a maximal converged eigenstate with eigenenergies $\epsilon_{k}\in \left[\epsilon_{\mathrm{GS}}=-2.125,\epsilon_{T}=1.841\right]$, for three values of the system size j=30, 60, 100. As discussed in Section 4 and in Figure 2, thermalization requires the convergence of the infinite-time average of a few-body observable towards the microcanonical average in the thermodynamic limit, so scaling analysis needs to be performed. As seen in Figure 5a, the patterns at the low energies of the regular region of the Dicke model do not disappear as the system size increases, but in the chaotic region, the fluctuations clearly shrink as the system size increases. This behavior is quantified in Figure 5b, where we present the deviation of the entanglement entropy from the microcanonical average, as computed in Equation (Equation 15) for the three system sizes j=30,60,100. The fluctuations decay to values close to zero for energies above ϵ≈−1.2, indicating the transition to a region where the majority of the eigenstates are chaotic. In the inset of Figure 5b, we present the extremal fluctuations calculated with Equation (Equation 16) for the chaotic energy interval only, ϵ∈[−0.5,1.841]. We avoid the regular region, where the changes in the extremal values are abrupt. The inset confirms the decrease in fluctuations with the increase in system size.

By comparing Figure 5b with Figure 3c,d, we observe that the deviation from the microcanonical average of the von Neumann entanglement entropy decreases to zero more abruptly as compared to the the number of photons and excited atoms. This different behavior between entropies and expectation values of observables is not well understood yet and motivates future work.

### 5.2. Results for the Shannon Entropy

We proceed with the analysis of the structure of the eigenstates of the Dicke model making use now of the basis-dependent Shannon entropy. In Figure 6a, we plot the Peres lattice of the Shannon entropy for the eigenstates of the Dicke model in the Fock and efficient bases. The eigenstates range from the ground state until a maximal converged eigenstate with eigenenergies $\epsilon_{k}\in \left[\epsilon_{\mathrm{GS}}=-2.125,\epsilon_{T}=2.356\right]$. The inset in Figure 6a contains the same data, but shows the exponential of the entropy. The values of the Shannon entropy computed in the Fock basis are larger than in the efficient basis. For the Fock basis, the entropy grows unboundedly with energy, while for the efficient basis, $S_{\mathrm{Sh}}$ saturates at high energies. These features make evident the advantages of using the efficient basis, since it requires fewer basis states to build a given eigenstate than what is needed by the Fock basis. While in Figure 6, our system size was restricted to j=30, when we employ the efficient basis, the same computational resources allow us to go up to j=100, as used in Figure 2, Figure 5 and Figure 7 below.

The analysis of the fluctuations of the Shannon entropy in Figure 6b and its inset shows that, similarly to what we observed for the entanglement entropy in Figure 5, they approach zero in the chaotic region. The results are comparable for both the Fock and efficient bases. In what follows, we examine the fluctuations of the Shannon entropy for different system sizes obtained for eigenstates written in the efficient basis only.

In Figure 7a, we plot the Peres lattice of the Shannon entropy for the eigenstates of the Dicke model written in the efficient basis. These eigenstates are contained in the energy interval $\epsilon_{k}\in \left[\epsilon_{\mathrm{GS}}=-2.125,\epsilon_{T}=1.841\right]$ for the same values of the system size which were considered previously, j=30,60,100. Similarly to what we observe for the entanglement entropy in Figure 5a, the regular patterns and large fluctuations are visible at low energies, where we do not expect the validity of ETH. The fluctuations decrease as we move to high energies and as the system size is increased. This behavior is quantified with the deviations from the microcanonical average shown in Figure 7b and its inset. The decay of $\Delta^{\mathrm{mic}}\left[S_{\mathrm{En}}\right]$ to values close to zero for the entanglement in Figure 5b is more abrupt than what we see for $\Delta^{\mathrm{mic}}\left[S_{\mathrm{Sh}}\right]$ in Figure 7b, but both cases signal the transition to chaos.

We close this section mentioning that the regular patterns seen at low energies in Figure 2 and Figure 5, Figure 6 and Figure 7 reflect the existence of families of periodic orbits, which were studied in previous works [61,62,106]. We also note that for high energies, the behavior of the Shannon entropy computed in the efficient basis and of the von Neumann entanglement entropy are similar in the sense that both measures nearly saturate. In contrast, the Shannon entropy in the Fock basis grows rapidly with energy, because of the infinite bosonic Hilbert subspace of the Dicke model. The modified bosonic sector of the efficient basis is responsible for the smaller values of the Shannon entropy when compared to the values for the Fock basis.

## 6. Conclusions

In this work, we confirmed the validity of the ETH in the chaotic region of the Dicke model. This was done by analyzing the diagonal and off-diagonal elements of the number of photons and the number of excited atoms for different system sizes. We corroborated that the validity of the ETH stems from the presence of chaotic eigenstates, which we showed by analyzing their components and measures of entanglement and delocalization.

The Shannon entropy, used to quantify the level of delocalization of the eigenstates in a given basis, made evident the advantages of using the efficient basis over the Fock basis. For high energies, the first leads to a slower growth of the entropy than the Fock basis, allowing us to reach converged states for larger system sizes than accessible with the Fock basis.

## Figures and Tables

**Figure 1 entropy-25-00008-f001:**
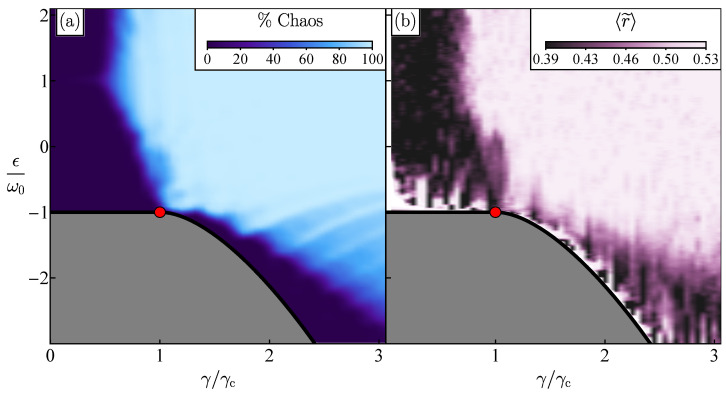
Panel (**a**): Map of the percentage of chaos for the classical trajectories as a function of the excitation energy ϵ and the coupling strength γ. Panel (**b**): Map of the average values of the ratio of consecutive energy levels $\langle \tilde{r}\rangle$ (see Equation (Equation 8)) obtained for moving windows of eigenenergies, which contain approximately 1000 energy levels. The low energy regions contain few energy levels and were averaged using windows of eigenenergies with the available energy levels. Because of this, the fluctuations of $\langle \tilde{r}\rangle$ are more pronounced in these regions. In both panels (**a**) and (**b**), the red dot represents the critical point for the transition from the normal to the superradiant phase for the ground state. The system size in panel (**b**) is j=100.

**Figure 2 entropy-25-00008-f002:**
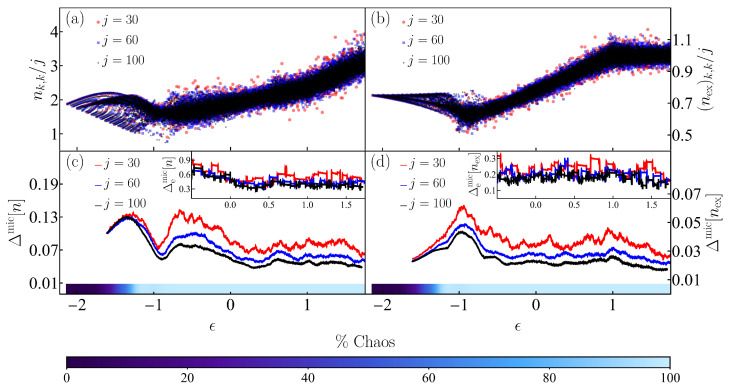
Panels (**a**) and (**b**): Peres lattice of the eigenstate expectation values of the number of photons $n_{k,k}$ (**a**) and the number of excited atoms ${\left(n_{\mathrm{ex}}\right)}_{k,k}$ (**b**) scaled to the system size *j* for eigenstates $|E_{k}\rangle$ with positive parity of the Dicke model. Panels (**c**) and (**d**): Deviations of the expectation values of the same observables $n_{k,k}$ (**c**) and ${\left(n_{\mathrm{ex}}\right)}_{k,k}$ (**d**) with respect to their microcanonical value (see Equation (Equation 15)). The insets in (**c**) and (**d**) show the extremal deviations of the respective quantities in the chaotic energy regime (see Equation (Equation 16)). The color scales contained within the panels (**c**) and (**d**) correspond to the values of the percentage of classical chaos shown at the bottom in the numbered color scale. The system size in all panels (**a**–**d**) is indicated with a given color for three values j=30,60,100.

**Figure 3 entropy-25-00008-f003:**
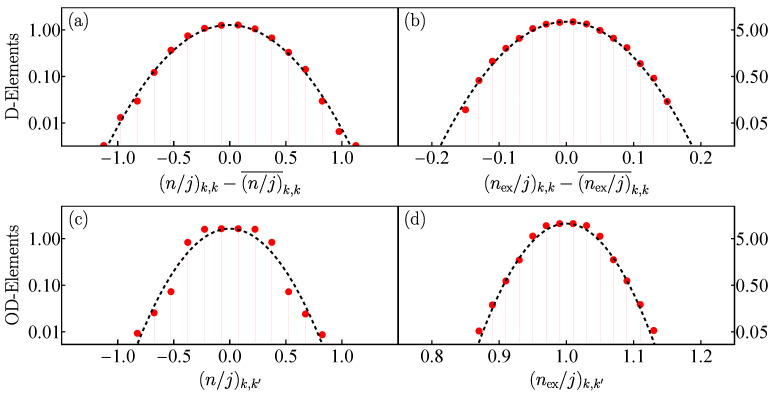
Panels (**a**) and (**b**): Statistical distribution (red solid dots) in semi-logarithmic scale of the diagonal matrix elements of number of photons $n_{k,k}$ (**a**) and excited atoms ${\left(n_{\mathrm{ex}}\right)}_{k,k}$ (**b**) scaled to the system size *j* for eigenstates $|E_{k}\rangle$ with positive parity of the Dicke model contained in the chaotic energy interval $\epsilon_{k}\in \left(0.5,1.0\right)$. The term ${\bar{O}}_{k,k}$ represents the average of the expectation value $O_{k,k}$ in the same energy region for each operator $\hat{n}$ and ${\hat{n}}_{\mathrm{ex}}$. The black dashed line depicts a Gaussian fit. Panels (**c**) and (**d**): The same as panels (**a**) and (**b**) for the off-diagonal matrix elements of the same observables $n_{k,k^{\prime }}$ (**c**) and ${\left(n_{\mathrm{ex}}\right)}_{k,k^{\prime }}$ (**d**). The system size in all panels (**a**–**d**) is j=30.

**Figure 4 entropy-25-00008-f004:**
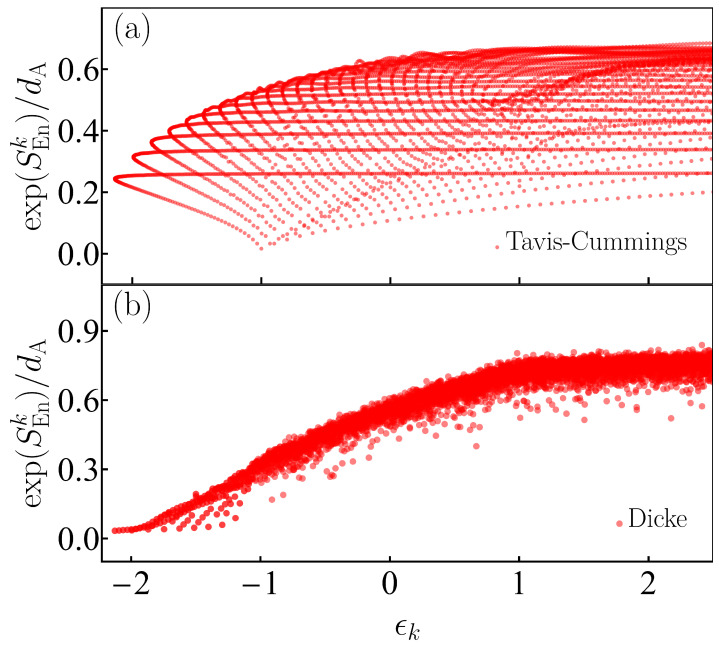
Panels (**a**) and (**b**): Peres lattice of the exponential of the von Neumann entanglement entropy $S_{\mathrm{En}}^{k}$ (see Equation (Equation 19)) scaled to the atomic Hilbert-space dimension $d_{A}=2j+1$ for eigenstates $|E_{k}\rangle$ from both parity sectors of the integrable Tavis–Cummings model (**a**) (see Equation (Equation 44)) and the nonintegrable Dicke model (**b**) (see Equation (Equation 2)). The system size in both panels (**a**) and (**b**) is j=30.

**Figure 5 entropy-25-00008-f005:**
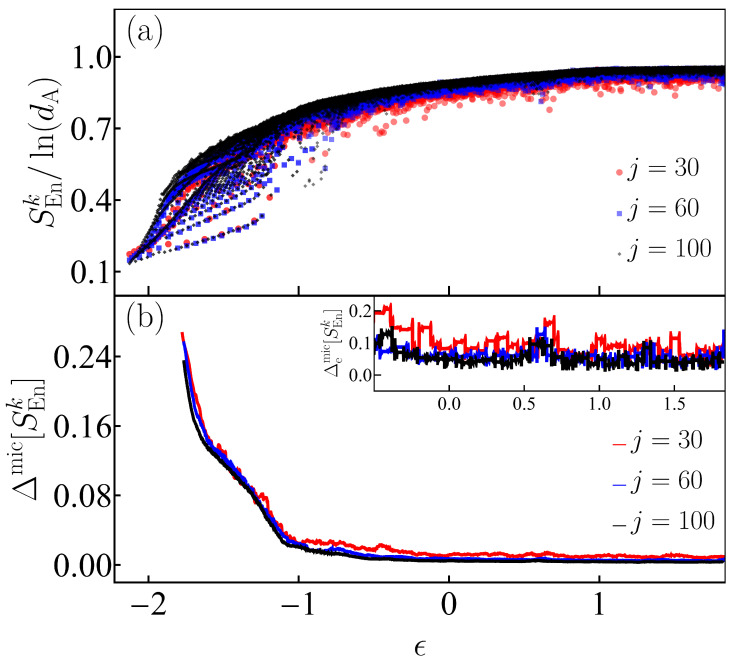
Panel (**a**): Peres lattice of the von Neumann entanglement entropy $S_{\mathrm{En}}^{k}$ scaled to the atomic Hilbert-space dimension $d_{A}=2j+1$ for eigenstates $|E_{k}\rangle$ from both parity sectors of the Dicke model. Panel (**b**): Deviations of the von Neumann entanglement entropy with respect to its microcanonical value, computed with Equation (Equation 15). The inset shows the extremal deviations in the chaotic energy regime computed with Equation (Equation 16). The system size in both panels (**a**) and (**b**) is indicated with a given color for three values j=30,60,100.

**Figure 6 entropy-25-00008-f006:**
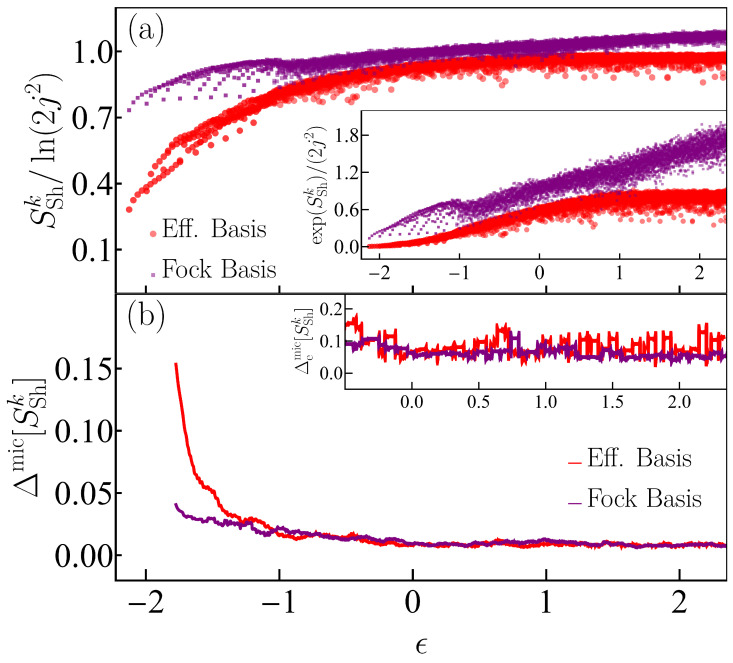
Panel (**a**): Peres lattices of the Shannon entropy $S_{\mathrm{Sn}}^{k}$ (see Equation (Equation 21)) scaled to the system-size dependence of the density of states, $\nu \left(\epsilon \right)\propto 2j^{2}$ (see Appendix B), for eigenstates $|E_{k}\rangle$ from both parity sectors of the Dicke model written in the efficient basis (red dots) and the Fock basis (purple dots). The inset shows the exponential values of the Shannon entropy for both bases. Panel (**b**): Deviation of Shannon entropy with respect to its microcanonical value (see Equation (Equation 15)). The inset shows the extremal deviation of the same quantity in the chaotic energy regime (see Equation (Equation 16)). The system size in both panels (**a**) and (**b**) is j=30.

**Figure 7 entropy-25-00008-f007:**
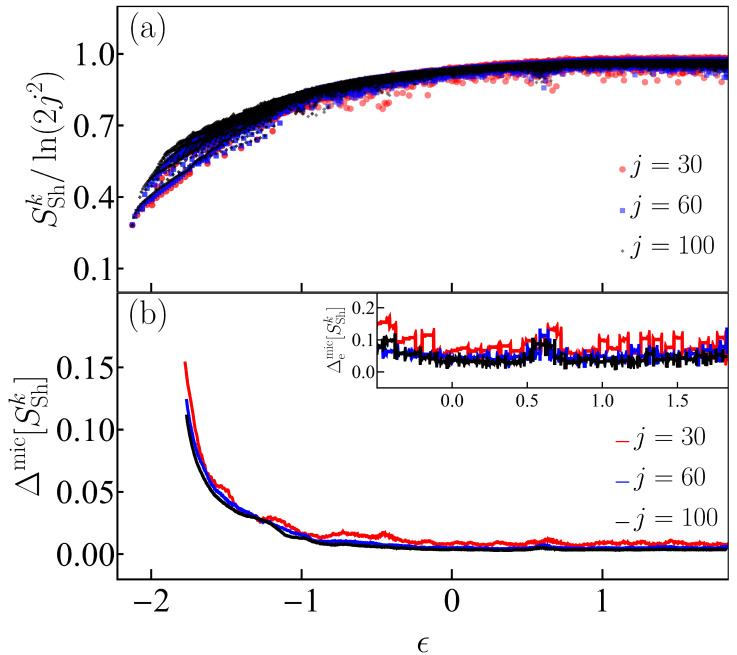
Panel (**a**): Peres lattice of Shannon entropy $S_{\mathrm{Sh}}^{k}$ scaled to the system-size dependence of the density of states $\nu \left(\epsilon \right)\propto 2j^{2}$ for eigenstates $|E_{k}\rangle$ from both parity sectors of the Dicke model. The Shannon entropy for these eigenstates was built in the efficient basis (see Section A.2). Panel (**b**): Deviations of the Shannon entropy with respect to its microcanonical value, computed with Equation (Equation 15). The inset shows the extremal deviations in the chaotic energy regime computed with Equation (Equation 16). The system size in both panels (**a**) and (**b**) is indicated with a given color for three values j=30,60,100.

## Data Availability

All the data that support the results and plots showed within this work are available from the corresponding authors upon request.

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
