# Peer review of "Chaos and Thermalization in the Spin-Boson Dicke Model"

_entropy, 2022, doi:10.3390/e25010008_

Round 1

Reviewer 1 Report

This is an interesting and important investigation of thermalization in a quantum-mechanical system with a classical limit with a mixed phase space. The research is timely and the results (particularly on the best choice of basis for models with an unbounded Hilbert space) are likely to influence future work. The paper is well written and presented. I strongly recommend publication.

Author Response

We thank the Referee for finding our research timely, our results likely to influence future work, our work well written and presented, and for strongly recommend publication as it stands.

Reviewer 2 Report

Referee report on Villasenor et al.:„ Chaos and Thermalization in the Spin-Boson Dicke Model”, entropy-2065671

This paper presents results on the “Peres lattice” for the spin-boson Dicke model. The idea is to probe the ETH hypothesis by checking on the smoothness of the Peres lattice for expectation values such as number of photons, of excited atoms, or the entanglement entropy. As quantitative values the authors use the deviation from the microcanonical average for the same variable. The authors observe that this deviation rapidly decreases to (near) zero for the entropies studied (fig. 5, 7) when the system transits to the chaotic regime but not nearly as pronounced for atom or photon related observables (fig. 3). Remarkably, the authors appear not to discuss this interesting difference which calls for an explanation.

Overall, the paper is quite well-written and informative. The introduction gives valuable insights and a brief history of the ETH. While one may take issue with a few statements there, it is a nice primer to newcomers in this field. This paper fits very well the occasion. In conclusion, a modified version with a somewhat expanded physics discussion should be included into this special issue.

Author Response

We thank the Referee for finding our work well-written and informative. Next, we reply point-by-point each one of the Referee’s comments and explain the changes that they moti- vated to our revised version.

Point-by-point reply:

• “This paper presents results on the “Peres lattice” for the spin-boson Dicke model. The idea is to probe the ETH hypothesis by checking on the smoothness of the Peres lattice for expectation values such as number of photons, of excited atoms, or the entanglement entropy. As quantitative values the authors use the deviation from the microcanonical average for the same variable. The authors observe that this deviation rapidly decreases to (near) zero for the entropies studied (fig. 5, 7) when the system transits to the chaotic regime but not nearly as pronounced for atom or photon related observables (fig. 3). Remarkably, the authors appear not to discuss this interesting difference which calls for an explanation.”

Reply. We thank very much Referee 2 for this very pertinent remark. We do not have an answer to this interesting feature between entropies and expectations values of observables in the Dicke model; nevertheless, this observation motivates us to keep researching for future work. We have added a little paragraph rephrasing this answer in the manuscript.

Reviewer 3 Report

See attachment

Author Response

We thank the Referee for thinking that our work is very well written and each step in the analysis is carefully explained. We thank the Referee also for suggesting publication of the manuscript as it stands. Next, we address in detail the Referee’s minor points.

Point-by-point reply:

• “In the caption of Fig. 1, the authors should explain the mean of the white region in Fig. 1 (b).”

Reply. We thank the Referee for this observation. We have modified Fig. 1 in order to avoid the white region in panel (b).

• “The reduced density matrix in the atomic sector in Eq. (18) is wrong, because it only

gives the diagonal elements, but ρA also has non-diagonal elements.”

Reply. We thank the Referee for this important remark. We have corrected Eq. (18) in the manuscript.